# Investigating the impact of the COVID-19 pandemic on pediatric adenovirus infection patterns: a retrospective analysis

Yun Gu,[1] Xin Zhang,[1] Lunjing Yan,[1] Yihan Wang,[1] Shenghao Hua[1]

**ABSTRACT** This study aimed to compare the epidemiological features and clinical manifestations of adenovirus (ADV)-associated pediatric respiratory tract infections during and after the lockdown due to coronavirus disease 2019 (COVID‐19) pandemic to explore potential immunity debt. A retrospective analysis was conducted to evaluate ADV detection rates, co-infection rates, serotype distribution, age distribution of infected children, proportions of mild, moderate, and severe ADV pneumonia cases, and relevant laboratory indicators (e.g., white blood cell count, granulocyte-to-lymphocyte ratio, high-sensitivity C-reactive protein, and lactate dehydrogenase) during and after the lockdown. Post-lockdown ADV detection and co-infection rates were significantly higher than those during the lockdown ($P < 0.05$). The predominant co-infection pathogens shifted from rhinovirus and bocavirus to rhinovirus and *Mycoplasma pneumoniae*. A marked increase in ADV detection was observed from October to December post-lockdown ($P < 0.05$). The age distribution of infected children also changed, with notable rises in cases aged 4–6 and >6 years. The dominant ADV serotype shifted from type 1 pre-pandemic to type 3 post-lockdown. The proportion of severe ADV pneumonia cases increased significantly ($P < 0.05$), although laboratory indicators showed no significant differences. Following the COVID-19 pandemic, significant alterations have emerged in the clinical and epidemiological profiles of pediatric ADV-associated respiratory tract infections. The observed rise in severe ADV pneumonia cases may correlate with pandemic-associated immunity debt in children, underscoring the necessity to reevaluate prevention and therapeutic strategies to mitigate potential escalations in critical illnesses and evolving infection patterns.

**IMPORTANCE** The coronavirus disease 2019 pandemic has significantly altered the epidemiology of adenoviral (ADV) infections in children, revealing phenomena such as "immunity debt." A comparative analysis of ADV detection rates, co-infection frequencies, age distributions, subtype shifts, and clinical manifestations during and after the lockdown indicates increased detection rates, a shift from type 1 to type 3, and a rise in severe cases of ADV pneumonia, necessitating reevaluation of public health strategies for pediatric viral infections.

**KEYWORDS** COVID-19 pandemic, adenovirus, pediatric respiratory tract infections, clinical symptoms

The coronavirus disease 2019 (COVID-19) pandemic, caused by the novel coronavirus SARS-CoV-2, has profoundly impacted public health worldwide, leading to significant changes in the epidemiology of various infectious diseases, including adenovirus (ADV) infections in children (1, 2). The emergence of SARS-CoV-2 has disrupted the typical patterns of viral infections, potentially resulting in what is termed "immunity debt," a phenomenon where the interruption of routine infections may lead to increased susceptibility to other pathogens once public health measures are

**Peer Reviewer** Wangqi Du, Taixing People's Hospital, Taixing, Jiangsu, China

Address correspondence to Shenghao Hua, sheng_hao1988@163.com.

The authors declare no conflict of interest

See the funding table on p. 9.

relaxed (3). Although the COVID-19 pandemic began in 2020, most regions of China, including Suzhou, did not experience widespread transmission until the end of 2022. This delay was largely due to the lockdown measures implemented to control the virus's spread (4). ADVs are categorized into various serotypes, with types 1, 2, and 3 being commonly associated with respiratory illnesses in children (5). Current research on ADV epidemiology has primarily focused on detection rate comparisons during and after the lockdown, without comprehensive investigation into subtype specificity, co-infection patterns, and associated clinical manifestations (6, 7). To investigate whether pediatric ADV infections exhibited analogous "immunity debt" phenomena after lockdown, we conducted a comparative analysis of the following parameters during and after the lockdown: ADV detection rates, co-infection frequency, viral subtype distribution, age-specific infection patterns, clinical severity stratification of adenoviral pneumonia cases (mild/moderate/severe), and relevant laboratory biomarkers.

## MATERIALS AND METHODS

### Study subjects

A total of 21,918 hospitalized children who underwent respiratory pathogen testing using sputum specimens at the Children's Hospital of Soochow University from 2022 to 2023 were enrolled. All children signed informed consent before admission, agreeing to take part in this clinical study. The nucleic acid samples of the sputum from children who tested positive for ADV will be stored at −80℃. From this cohort, 51 ADV nucleic acid-positive samples were selected for genotyping analysis based on the following criteria: (i) absence of severe underlying diseases and (ii) presentation with classical respiratory infection symptoms. Clinical data and laboratory parameters were retrospectively collected from 112 ADV pneumonia cases in 2022 and 175 cases in 2023. Disease severity was classified as mild, moderate, or severe according to the Guidelines for Diagnosis and Treatment of Pediatric Adenoviral Pneumonia (2019 Edition) and WHO criteria for pediatric pneumonia severity (Table 1).

Exclusion criteria were (i) incomplete clinical data and (ii) comorbid hematologic disorders, malignancies, or congenital heart diseases.

TABLE 1  Classification criteria for the severity of pediatric adenovirus pneumonia

| Severity | Clinical criteria |
|---|---|
| Mild | 1. Upper respiratory symptoms: fever (38–39℃), cough, rhinorrhea, pharyngodynia, hoarseness ± conjunctivitis (hyperemia, exudates); |
| | 2. Systemic manifestations: anorexia, fatigue, myalgia; and |
| | 3. Gastrointestinal involvement: diarrhea, vomiting, abdominal pain (subgroup). |
| Moderate | 1. Persistent fever: 39–40℃ lasting >5 days with poor antipyretic response; |
| | 2. Progressive respiratory distress: productive cough/wheezing, exertional dyspnea, audible rales/rhonchi; and |
| | 3. Radiographic findings: unilateral/bilateral patchy infiltrates ± coalescence. |
| Severe | 1. Critical respiratory failure: dyspnea with nasal flaring/chest retractions, hypoxemia ($PaO_2/FiO_2 < 300$), requiring $O_2$/mechanical ventilation; |
| | 2. Hyperpyrexia (>40℃) with neurological sequelae: lethargy, irritability, or altered consciousness; and |
| | 3. Multisystem complications: ARDS, pleural effusion, myocarditis, encephalitis/meningitis (seizures), hepatic dysfunction, coagulopathy, septic shock, or MOF. |

**TABLE 2** Comparison of ADV detection and co-infection rates during and after the lockdown [*n*/total cases (%)]

| Rate | During the lockdown | After the lockdown | *P* value |
|------|--------------------|--------------------|-----------|
| ADV detection rate | 167/7,689 (2.17) | 490/14,299 (3.44) | <0.0001 |
| ADV co-infection rate | 75/167 (44.91) | 310/490 (63.27) | <0.0001 |

## Specimen collection

Sputum samples were obtained within 24 h of admission via sterile suction catheter (inserted 7–8 cm nasally) under negative pressure. Secretions (1–2 mL) were transferred to 0.9% NaCl-containing tubes, homogenized, and immediately transported for analysis.

## Pathogen detection

Extraction of nucleic acid with an automated extraction system (Jiangsu Shuoshi Biotech). Detection of 13 respiratory pathogens (RSV, FluA, FluA-H1N1, FluA-H3N2, FluB, HPIV, ADV, MP, Ch, Boca, HMPV, HRV, and HCOV) with Ningbo HealthGene 13-plex RT-PCR Kit. ADV genotyping used type-specific PCR (Ningbo HealthGene) for serotypes 1–5, 7, 14, 21, 37, 40, 41, and 55.

## Statistical analysis

Categorical variables were expressed as frequencies (%) and analyzed by chi-square test using GraphPad Prism 5.0. Statistical significance was defined as $P < 0.05$.

## RESULTS

### Comparative analysis of pediatric ADV detection and co-infection rates during and after the lockdown

During the nationwide implementation of routine containment measures in 2022, respiratory specimens from 7,689 cases were tested, identifying 167 ADV-positive cases. Among these, 92 (55.1%) were single ADV infections, while 75 (44.9%) involved co-infections with other respiratory pathogens. Following the discontinuation of containment measures on 5 December 2022, testing of 14,229 specimens in 2023 revealed 490 ADV-positive cases, demonstrating a significantly higher detection rate compared to

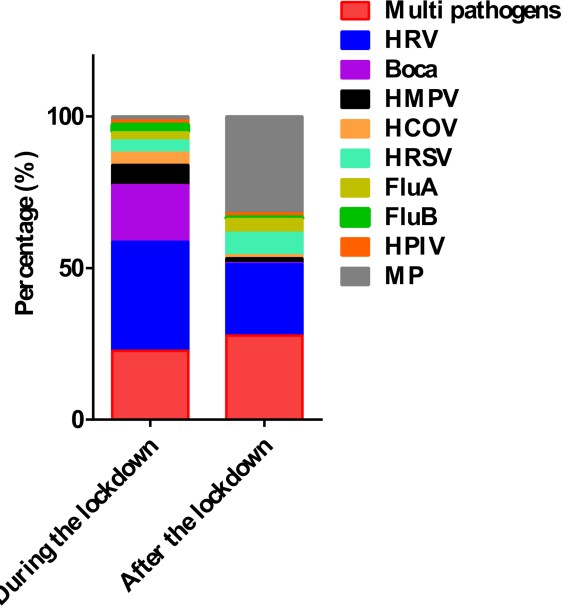

**FIG 1** The proportional distribution of co-infection pathogens with ADV before and after the pandemic.

2022 (3.44% vs 2.17%, *P* < 0.0001). Co-infection rates also increased markedly, with 310 out of 490 (63.3%) cases involving pathogen co-detection versus 75 out of 167 (44.9%) in 2022 (*P* < 0.0001, Table 2).

ADV co-infections predominantly involved single-pathogen combinations. In 2022, 59 cases (78.67%) exhibited single co-infections, while 16 cases (21.33%) showed multiple co-infections (≥2 pathogens). By 2023, single co-infections rose to 221 cases (71.29%), with 89 cases (28.71%) involving multiple pathogens. The proportional distribution of co-detected pathogens with ADV is illustrated in Fig. 1, with multiple pathogen (>2 pathogens) combinations and case numbers provided in Fig. 2.

**A**

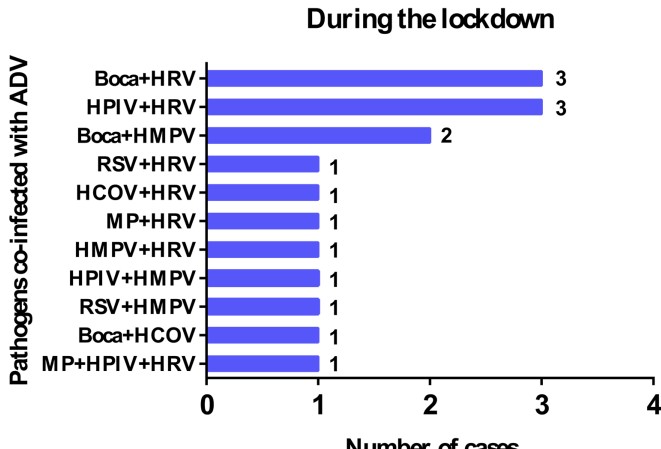

**B**

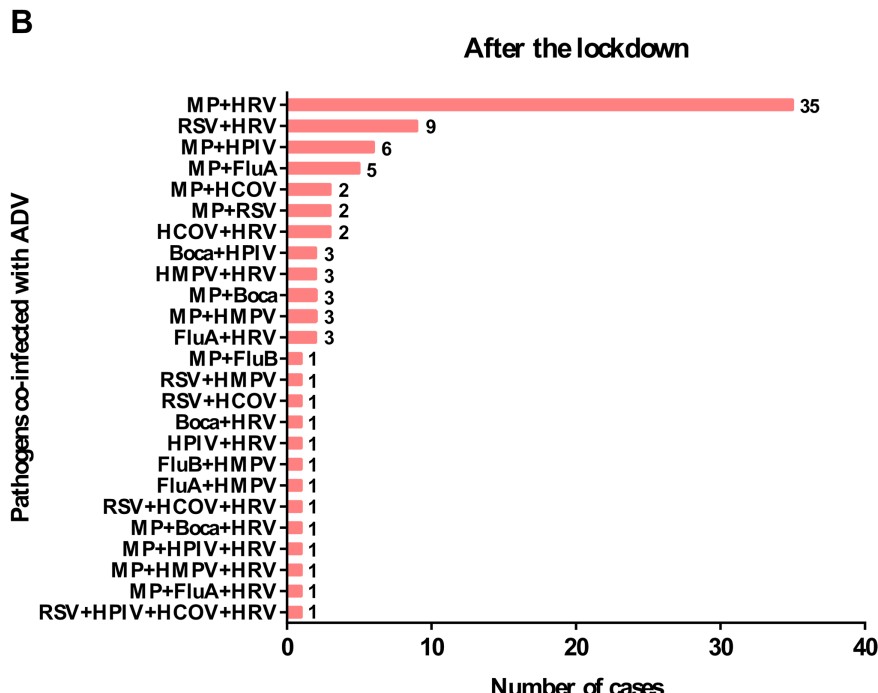

**FIG 2** Pathogens co-infected with ADV and case numbers during (A) and after (B) the lockdown.

**TABLE 3** Monthly comparison of ADV detection rates during and after the lockdown [*n*/total cases (%)]

| Month | During the lockdown | After the lockdown | *P* value |
|---|---|---|---|
| January | 19/955 (1.99) | 4/250 (1.60) | >0.9999 |
| February | 10/387 (2.58) | 15/398 (3.77) | 0.4180 |
| March | 10/377 (2.65) | 10/945 (1.06) | 0.0438 |
| April | 5/142 (3.52) | 22/1,066 (2.06) | 0.2360 |
| May | 5/223 (2.24) | 30/1,376 (2.18) | >0.9999 |
| June | 8/502 (1.59) | 25/1,303 (1.92) | 0.8446 |
| July | 17/631 (2.69) | 9/1,284 (0.70) | 0.0011 |
| August | 20/827 (2.42) | 28/1,380 (2.03) | 0.5494 |
| September | 18/905 (1.99) | 36/1,430 (2.52) | 0.4805 |
| October | 14/830 (1.69) | 72/1,596 (4.51) | 0.0003 |
| November | 21/1,102 (1.91) | 86/1,579 (5.45) | <0.0001 |
| December | 20/808 (2.48) | 153/1,622 (9.43) | <0.0001 |

## Comparative analysis of monthly ADV detection rates in children during and after the lockdown

During the initial 9-month post-lockdown period, ADV detection rates remained comparable to levels during the lockdown, with only March and July exhibiting significantly lower rates (*P* < 0.05). This temporal reduction potentially correlates with seasonal epidemics of flu in March and MP in July. From October onward, post-lockdown ADV detection rates demonstrated statistically significant increases compared to pre-pandemic baselines (*P* < 0.05), peaking at a 9.43% positivity rate in December (Table 3, Fig. 3).

## Comparative analysis of age distribution among ADV-positive pediatric cases during and after the lockdown

The 1–3 years age group predominated among ADV-positive cases during the lockdown period (52.69%, 88/167), with the proportion decreasing significantly to 29.80% (146/490) post-lockdown (*P* < 0.05). Concurrently, the 4–6 years cohort increased from 25.15% (42/167) to 37.35% (183/490), while the >6 years group rose from 13.17%

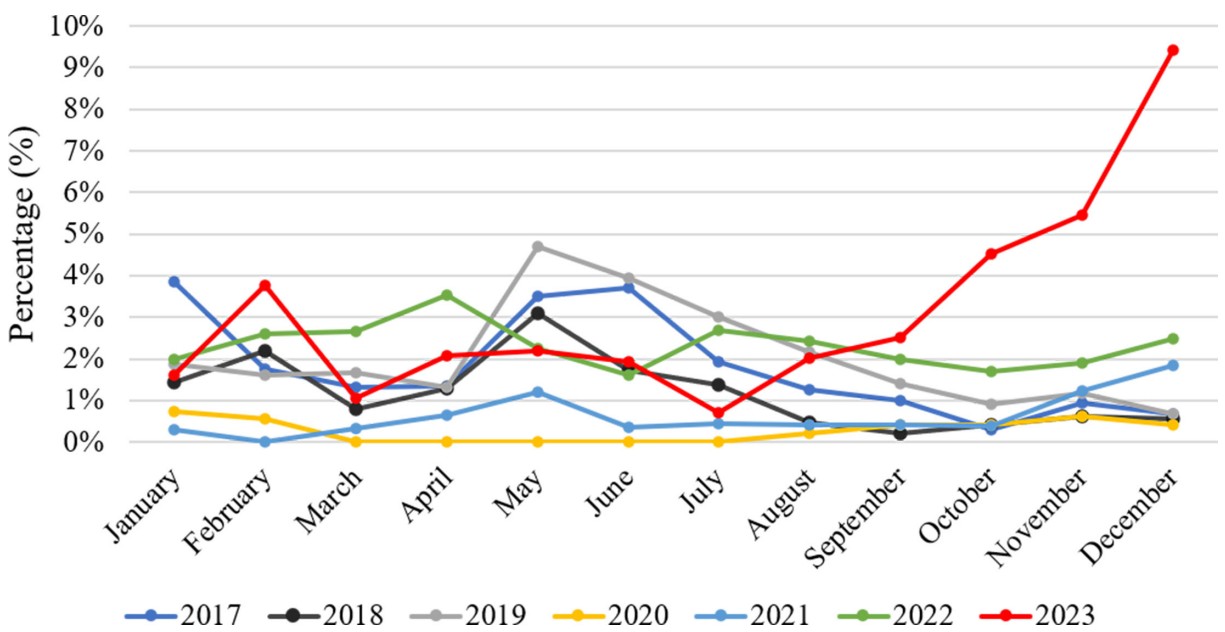

**FIG 3** Monthly ADV detection rates from 2017 to 2023.

**TABLE 4** Age-specific distribution of ADV-positive cases during and after the lockdown [*n*/total cases (%)]

| Age (years) | During the lockdown | After the lockdown | *P* value |
| --- | --- | --- | --- |
| <1 | 15/167 (8.98%) | 26/490 (5.31%) | 0.0971 |
| 1–3 | 88/167 (52.69%) | 146/490 (29.80%) | <0.0001 |
| 4–6 | 42/167 (25.15%) | 183/490 (37.35%) | 0.0045 |
| >6 | 22/167 (13.17%) | 135/490 (27.55%) | 0.0001 |

(22/167) to 27.55% (135/490), both demonstrating statistically significant increases (*P* < 0.05, Table 4 and Fig. 4).

## Comparative analysis of ADV subtype distribution in children during and after the lockdown

During the lockdown, ADV subtypes were predominated by type 1 (46.15%, 12/26), with additional detection of types 2 (26.92%, 7/26), 3 (23.08%, 6/26), and 55 (3.85%, 1/26). In contrast, the post-lockdown period demonstrated a significant subtype shift, with type 3 accounting for 84% (21/25) of cases, accompanied by minimal detection of types 1 (4%, 1/25), 2 (8%, 2/25), and 5 (4%, 1/25), as shown in Fig. 5.

## Comparative analysis of clinical manifestations and laboratory parameters in children with ADV pneumonia during and after the lockdown

Clinical data and laboratory parameters (white blood cell count, neutrophil-to-lymphocyte ratio, high-sensitivity C-reactive protein, and lactate dehydrogenase) were retrospectively analyzed in 112 ADV pneumonia cases (2022) and 175 cases (2023). Stratification by severity (mild/moderate/severe) according to the Guidelines for Diagnosis and Treatment of Pediatric Adenoviral Pneumonia (2019 Edition) and WHO criteria revealed a significant increase in severe cases post-pandemic (18.29% vs 7.14%, *P* < 0.05), while mild/moderate proportions remained statistically comparable (Table 5).

Comparative analysis of laboratory parameters across severity subgroups demonstrated significantly reduced WBC counts exclusively in mild (*P* = 0.013) and severe (*P* = 0.007) cases post-lockdown, with no intergroup differences in other biomarkers, indicating preserved pathophysiological severity patterns across pandemic phases (Fig. 6).

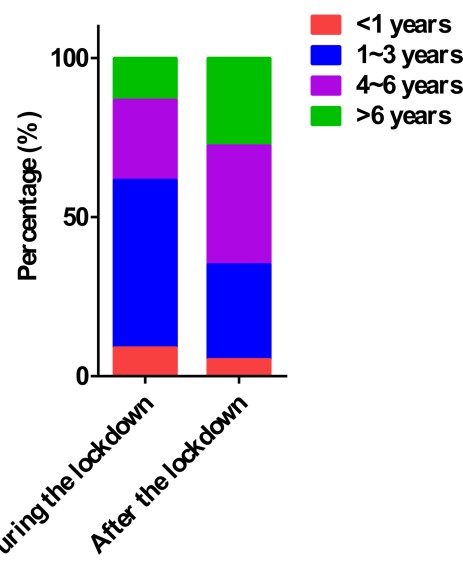

**FIG 4** Age distribution of ADV-positive pediatric cases during and after the lockdown.

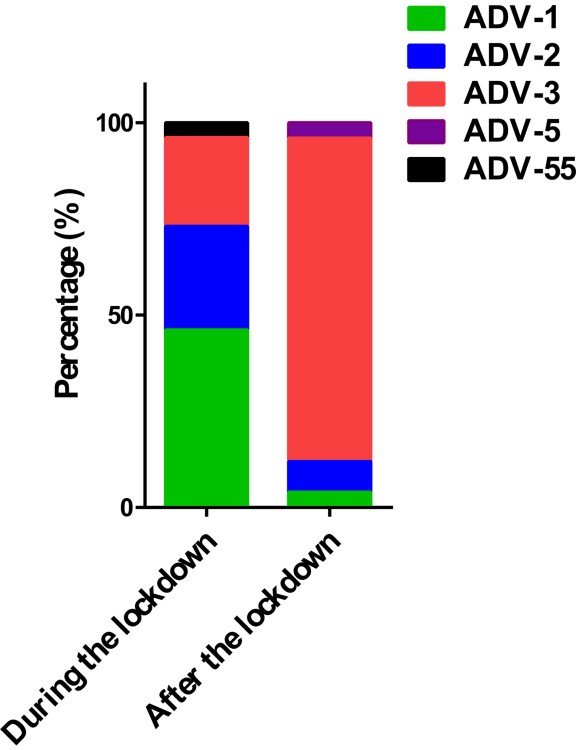

**FIG 5** ADV subtype distribution in children during and after the lockdown.

## DISCUSSION

The COVID-19 pandemic has reshaped the global landscape of respiratory infections, with emerging evidence concerning shifts in ADV epidemiology among children (1). ADV remains a critical etiological agent of pediatric respiratory infections, particularly in children under five years old, who are susceptible to severe complications such as bronchiolitis, pneumonia, and acute respiratory distress syndrome (8, 9). Pre-pandemic studies highlighted ADV as the second most prevalent non-influenza respiratory virus, accounting for 5–10% of hospitalized cases (10). Post-COVID-19 surveillance, however, suggests altered circulation patterns, potentially linked to immunity debt from prolonged non-pharmaceutical interventions (NPIs), underscoring the need to re-evaluate its clinical and epidemiological significance in the post-pandemic era.

Several factors may drive ADV's evolving prevalence and severity (11, 12). First, immunity debt resulting from reduced exposure to ADVs during lockdowns likely increased pediatric susceptibility upon relaxation of NPIs, triggering atypical seasonal surges. Second, viral interference dynamics shifted as SARS-CoV-2, influenza, and RSV re-emerged, potentially altering ADV co-infection rates and clinical outcomes. Third, children with prior SARS-CoV-2 infection may demonstrate heightened susceptibility to adenoviral infections. Previous studies indicate that ADV subtype does not vary year to year, but rather follows a cyclical epidemic pattern (13, 14). Therefore, the epidemiology of ADV is dynamic, with shifts in predominant serotypes and infection patterns observed over time, influenced by environmental factors and population immunity. As a result, understanding the nuances of ADV infections, including their clinical manifestations and associated complications, remains a crucial area of research in pediatric infectious diseases.

In this study, we have identified significant changes in the characteristics and clinical symptoms of ADV infections in children following the COVID-19 pandemic. The findings highlight a notable increase in ADV detection rates and mixed infections, particularly with the emergence of new pathogen combinations. This is a critical innovation in understanding the epidemiological shifts caused by the COVID-19 pandemic, as earlier

**TABLE 5** Severity stratification of ADV pneumonia cases during and after the lockdown [*n*/total cases (%)]

| Severity | During the lockdown | After the lockdown | *P* value |
|---|---|---|---|
| Mild | 36/112 (32.14) | 57/175 (32.57) | >0.9999 |
| Moderate | 67/112 (59.82) | 86/175 (49.14) | 0.0897 |
| Severe | 9/112 (8.04) | 32/175 (18.29) | 0.0158 |

studies primarily focused on individual infections without considering the broader implications of co-infections and population immunity debt (15). For instance, the shift in prevalent ADV types from types 1 to 3, along with the rise in severe cases of ADV pneumonia, suggests an evolving viral landscape that may be influenced by the immunity debt phenomenon observed in children post-pandemic (3). Evidence shows that the ADV subtypes 3 and 7 are more likely to cause severe pneumonia among children compared to the ADV subtypes 1, 2, and 6 (16, 17). A study from South Korea found that, following the COVID-19 pandemic, the ADV subtypes shifted from types 1 and 2 to type 3 both in pediatrics and adults (18). The shift to ADV type 3 was also seen following the COVID-19 pandemic in Jining, China (19). However, there are differences in the prevalence of the ADV subtypes common in various parts of the world (12). The increase in children aged four and older who tested positive for ADV after the COVID-19 pandemic might be linked to cross-infection events in schools. The implications of these findings extend to clinical practice and policy-making, emphasizing the need for reevaluation of current prevention and treatment strategies for pediatric viral infections. The observed changes in the age distribution of infected children also necessitate targeted public health interventions to address the unique vulnerabilities of these groups. These findings align with previous literature that suggests a resurgence of respiratory infections following periods of respiratory disease pandemic, underscoring the importance of adaptive healthcare responses in managing emerging infectious diseases.

While our study provides valuable insights, it is important to acknowledge certain limitations. The retrospective design may introduce biases related to data collection and interpretation, particularly in distinguishing between ADV and other respiratory

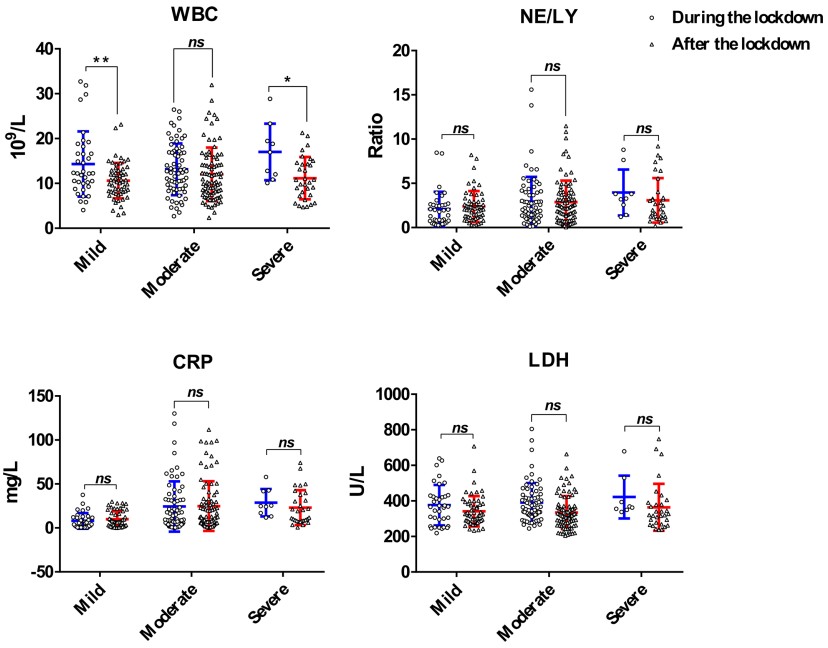

**FIG 6** Laboratory parameters across severity subgroups during and after the lockdown. *\*P* < 0.05; *\*\*P* < 0.01; CRP, C-reactive protein; LDH, lactate dehydrogenase; NE/LY, the ratio of neutrophils to lymphocytes; WBC, white blood cell.

pathogens in mixed infections. Moreover, the relatively short follow-up period may restrict our understanding of the long-term effects of these infections on children's health. Future studies should aim to include larger, multi-center cohorts to validate our findings and explore the underlying mechanisms that contribute to the observed changes in ADV infection dynamics post-pandemic. Addressing these limitations will enhance our understanding of pediatric respiratory infections and inform effective public health strategies moving forward.

In conclusion, the results of this study highlight a significant shift in the epidemiological characteristics of ADV infections in children following the COVID-19 pandemic. The increase in ADV detection rates, particularly among older age groups, and the rise in severe ADV pneumonia cases may suggest an emerging immunological debt phenomenon. These findings underscore the necessity for ongoing surveillance and reevaluation of public health strategies to mitigate the risks associated with future respiratory infections in children. A comprehensive understanding of the changing landscape of respiratory pathogens is essential for informing preventive measures and therapeutic approaches in pediatric populations.

## ACKNOWLEDGMENTS

Each author has made an important scientific contribution to the study. Y.G. and S.H. carried out the conception of the research, designed the research, and drafted the manuscript. Y.G. participated in collecting clinical data. X.Z., L.Y., and Y.W. collected samples and performed the test. S.H. performed the statistical analysis. All authors read and approved the final manuscript.

## AUTHOR AFFILIATION

[1]Department of Clinical Laboratory, Children's Hospital of Soochow University, Suzhou, Jiangsu, China

## AUTHOR ORCIDs

Shenghao Hua http://orcid.org/0009-0000-4422-9139

## FUNDING

| Funder | Grant(s) | Author(s) |
| --- | --- | --- |
| Suzhou Municipal Health Commission | KJXW2023025 | Xin Zhang |

## AUTHOR CONTRIBUTIONS

Yun Gu, Data curation, Writing – original draft | Xin Zhang, Funding acquisition, Project administration | Lunjing Yan, Project administration | Yihan Wang, Project administration.

## ETHICS APPROVAL

Our research has received approval from the Ethics Board at the Children's Hospital of Soochow University (2025CS135).

## ADDITIONAL FILES

The following material is available online.

Open Peer Review

**PEER REVIEW HISTORY (review-history.pdf).** An accounting of the reviewer comments and feedback.

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
