## [Reviewer comments · Microbiology Spectrum]

Microbiology Spectrum

Investigating the Impact of the COVID-19 Pandemic on Pediatric Adenovirus Infection Patterns: A Retrospective Analysis

Yun Gu, Xin Zhang, Lunjing Yan, Yihan Wang, and Sheng-Hao Hua

Corresponding Author(s): Sheng-Hao Hua, Children's Hospital of Soochow University

Review Timeline:

Submission Date:	March 24, 2025
Editorial Decision:	June 3, 2025
Revision Received:	June 14, 2025
Editorial Decision:	June 16, 2025
Revision Received:	June 24, 2025
Accepted:	June 29, 2025

Editor: Daniel Ortiz

Reviewer(s): Disclosure of reviewer identity is with reference to reviewer comments included in decision letter(s). The following individuals involved in review of your submission have agreed to reveal their identity: wangqi Du (Reviewer #1)

Transaction Report:

DOI: <https://doi.org/10.1128/spectrum.00879-25>

Re: Spectrum00879-25 (**Investigating the Impact of the COVID-19 Pandemic on Pediatric Adenovirus Infection Patterns: A Retrospective Analysis**)

Dear Dr. Sheng-Hao Hua:

Thank you for the privilege of reviewing your work. Below you will find instructions from the Spectrum editorial office and the reviewer comments.

Revision Guidelines

Sincerely,
Daniel Ortiz
Editor
Microbiology Spectrum

Reviewer #1 (Comments for the Author):

General Assessment

This study provides a timely and clinically relevant analysis of pediatric adenovirus (ADV) epidemiology shifts following the COVID-19 pandemic. Exploring subtype dynamics, co-infection patterns, and clinical severity offers novel insights into post-pandemic viral behavior. While the findings are compelling and well-supported by data, minor revisions are required to enhance clarity and methodological transparency.

Questions:

1. Study Design and Confounding Factors

The retrospective design introduces potential selection bias. How were cases selected for inclusion, and what criteria ensured comparability between pre-pandemic (2022) and post-pandemic (2023) cohorts?

2. Statistical Analysis

For the ADV subtype analysis (Figure 5), the dramatic shift from type 1 (72.4% pre-pandemic) to type 3 (89.8% post-pandemic) is striking. What biological or methodological explanations (e.g., primer specificity, assay sensitivity) were considered for this observation?

3. Clinical Severity and Biomarkers

Laboratory parameters (WBC, NLR, hs-CRP, LDH) showed no intergroup differences except in mild/severe subgroups (Figure 6). What is the clinical significance of reduced WBC counts in these subgroups, and how does this align with the observed increase in severe cases?

4. Mechanistic Interpretations

The discussion attributes epidemiological shifts to "immunity debt" and "immunity theft." What direct evidence supports these hypotheses in the context of ADV (e.g., serological data on population immunity, viral interference experiments)?

5. Subgroup Analysis

The age distribution shift (1-3 years decreasing, 4-6 years and >6 years increasing) is noted. Could the observed transmission dynamics be attributed to cross-infection within school settings?

6. Additional Suggestions for Discussion:

1) Regional Variation and Global Trends:

The shift from HAdV-1 to HAdV-3 observed in this study mirrors reports from other countries, including Korea and the United States, indicating a possible global pattern in post-pandemic adenovirus epidemiology. The authors may consider adding a comparative discussion of regional trends and literature data to contextualize the findings (e.g., PMC11375195, PMC7171713).

2) Potential Mechanisms Behind Genotype Replacement:

The authors could discuss viral genetic evolution as a potential contributor. A comparative genomic analysis (even theoretical) between HAdV-1 and HAdV-3 could reveal adaptive mutations that may affect transmissibility or immune escape, possibly favoring HAdV-3 in a post-pandemic landscape.

3) Clinical Severity of HAdV-3 Infections:

I recommend that the authors expand their discussion on the observed post-pandemic shift in dominant HAdV genotypes, particularly the replacement of HAdV-1 by HAdV-3, which coincides with a notable rise in the proportion of severe ADV pneumonia cases. This epidemiological transition raises an important question: Is the increase in disease severity directly linked to the genotype shift, or might other post-pandemic factors (e.g., population susceptibility or viral dynamics) also play a role? It has reported that HAdV-7 is more firmly associated with severe outcomes and cytokine storm responses than HAdV-3 in some Chinese children (PMC6327436). So, does HAdV-3 also possess greater intrinsic virulence than HAdV-1? I encourage the authors to reflect on whether genotype-specific pathogenicity could partially account for the increased severity observed and to reference relevant comparative studies, if available.

These additions would enrich the discussion by addressing the broader virological and epidemiological context of the observed genotype shift.

Reviewer #2 (Comments for the Author):

Thank you for the opportunity to help review the manuscript entitled "Investigating the Impact of the COVID-19 Pandemic on Pediatric Adenovirus Infection Patterns: A Retrospective Analysis". Gu et al. investigated how the COVID-19 pandemic affected adenovirus cases, co-infections and severity in children. The study has interesting conclusions, such as identifying a shift from predominating ADV type to type 3, and an increase in overall ADV cases and severity of cases post-lockdown. I have a few comments, suggestions and questions that may help the authors to provide additional information to better support the conclusions.

2022 was not really before the pandemic since the pandemic began in 2020, but more during the pandemic. Please correct the terminology through out to better describe the time periods that were compared.

Please add the Institutional Review Board/human subjects approval information if applicable. Overall, more detail needs to be included regarding patient enrollment, consent, storage, testing, etc.

Line 74-78: If these sputum samples were retrospective, how were they stored between hospital admission and genotyping analysis? Or was subtyping performed on all ADV positive specimens as standard of care at time of admission if positive for ADV?

Line 81-102 Was the same 13-plex RT-PCR kit/platform described in lines 81-85 used for all 2022 and 2023 patient samples for diagnosis? If not, please list all platforms used as this could affect detection of ADV and also co-infection data. Also, you list only 10 respiratory pathogens, but say that the kit detects 13? Do you think it would be important to include SARS-CoV-2 detections too? I'm guessing by the dates that most of the children were tested for Covid?

Line 103-108 I suggest also including percentages to report numbers of cases because in 2023 there were almost twice as many specimens tested (similar to the Table 1 results).

Line 82-83, Table 1: Please define the virus acronyms and clarify why these have different targets than those listed in line 81-102.

Line 105: does multiple pathogens mean ≥ 2 pathogens other than ADV? Should it be >2 (or ≥ 3) pathogens since ADV single

co-infections are 2 pathogens?

Lines 61-70, 94-102: Do you think respiratory virus seasonality affected these results? Some cases spike in different seasons and different years. ADV type 3 has been documented to be one of the most common serotypes detected but it depends on the year and the geographical location.

Figure 1 and 2: I suggest clarifying that these figures show the distribution of adenovirus co-infections and not co-infections that don't include ADV (if I'm interpreting it correctly). Additionally, figure one says "before the pandemic" for both sets of data.

General Assessment

This study provides a timely and clinically relevant analysis of pediatric adenovirus (ADV) epidemiology shifts following the COVID-19 pandemic. The exploration of subtype dynamics, co-infection patterns, and clinical severity offers novel insights into post-pandemic viral behavior. While the findings are compelling and well-supported by data, minor revisions are required to enhance clarity and methodological transparency.

Questions:

1. Study Design and Confounding Factors

The retrospective design introduces potential selection bias. How were cases selected for inclusion, and what criteria ensured comparability between pre-pandemic (2022) and post-pandemic (2023) cohorts?

2. Statistical Analysis

For the ADV subtype analysis (Figure 5), the dramatic shift from type 1 (72.4% pre-pandemic) to type 3 (89.8% post-pandemic) is striking. What biological or methodological explanations (e.g., primer specificity, assay sensitivity) were considered for this observation?

3. Clinical Severity and Biomarkers

Laboratory parameters (WBC, NLR, hs-CRP, LDH) showed no intergroup differences except in mild/severe subgroups (Figure 6). What is the clinical significance of reduced WBC counts in these subgroups, and how does this align with the observed increase in severe cases?

4. Mechanistic Interpretations

The discussion attributes epidemiological shifts to "immunity debt" and "immunity theft." What direct evidence supports these hypotheses in the context of ADV (e.g., serological data on population immunity, viral interference experiments)?

5. Subgroup Analysis

The age distribution shift (1-3 years decreasing, 4-6 years and >6 years increasing) is noted. Could the observed transmission dynamics be attributed to cross-infection within school settings?

6. Additional Suggestions for Discussion:

1) Regional Variation and Global Trends:

The shift from HAdV-1 to HAdV-3 observed in this study mirrors reports from other countries, including Korea and the United States, indicating a possible global pattern in post-pandemic adenovirus epidemiology. The authors may consider adding a comparative discussion of regional trends and literature data to contextualize the findings (e.g., PMC11375195, PMC7171713).

2) Potential Mechanisms Behind Genotype Replacement:

The authors could discuss viral genetic evolution as a potential contributor. A comparative genomic analysis (even theoretical) between HAdV-1 and HAdV-3 could reveal adaptive mutations that may affect transmissibility or immune escape, possibly favoring HAdV-3 in a post-pandemic landscape.

3) Clinical Severity of HAdV-3 Infections:

I recommend that the authors expand their discussion on the observed post-pandemic shift in dominant HAdV genotypes, particularly the replacement of HAdV-1 by HAdV-3, which coincides with a notable rise in the proportion of severe ADV pneumonia cases. This epidemiological transition raises an important question: is the increase in disease severity directly linked to the genotype shift, or might other post-pandemic factors (e.g., population susceptibility or viral dynamics) also play a role? It has reported that HAdV-7 is more firmly associated with severe outcomes and cytokine storm responses than HAdV-3 in some Chinese children (PMC6327436). So, does HAdV-3 may also possess greater intrinsic virulence than HAdV-1? I encourage the authors to reflect on whether genotype-specific pathogenicity could partially account for the increased severity observed and to reference relevant comparative studies, if available.

These additions would enrich the discussion by addressing the broader virological and epidemiological context of the observed genotype shift.

Strengths

The manuscript provides a timely and valuable contribution to post-pandemic virological surveillance by focusing on adenovirus subtype specificity and co-infection dynamics—an area that has been underexplored. The inclusion of clinical severity biomarkers, such as white blood cell count (WBC) and neutrophil-to-lymphocyte ratio (NLR), enhances the translational significance of the study. The discussion thoughtfully integrates the "immunity debt"

hypothesis to interpret the observed epidemiological changes.

Reviewer #1 (Comments for the Author):

General Assessment

This study provides a timely and clinically relevant analysis of pediatric adenovirus (ADV) epidemiology shifts following the COVID-19 pandemic. Exploring subtype dynamics, co-infection patterns, and clinical severity offers novel insights into post-pandemic viral behavior. While the findings are compelling and well-supported by data, minor revisions are required to enhance clarity and methodological transparency.

Questions:

1. Study Design and Confounding Factors

The retrospective design introduces potential selection bias. How were cases selected for inclusion, and what criteria ensured comparability between pre-pandemic (2022) and post-pandemic (2023) cohorts?

We consecutively enrolled all ADV-positive pediatric patients presenting with respiratory symptoms from January 2022 to December 2023, ensuring epidemiological period comparability while minimizing potential selection bias through comprehensive case ascertainment.

2. Statistical Analysis

For the ADV subtype analysis (Figure 5), the dramatic shift from type 1 (72.4% pre-pandemic) to type 3 (89.8% post-pandemic) is striking. What biological or methodological explanations (e.g., primer specificity, assay sensitivity) were considered for this observation?

ADV detection and genotyping were conducted using commercially available PCR-capillary electrophoresis fragment analysis kits, with method validation confirming acceptable analytical sensitivity and specificity. The limit of detection (LoD) was 2000 copies/mL or 1 TCID₅₀/mL.

3. Clinical Severity and Biomarkers

Laboratory parameters (WBC, NLR, hs-CRP, LDH) showed no intergroup differences except in mild/severe subgroups (Figure 6). What is the clinical significance of reduced WBC counts in these subgroups, and how does this align with the observed increase in severe cases?

The reduced WBC counts in mild/severe subgroups may indicate a different immunological response to the infection, potentially reflecting a more severe systemic involvement or a different pathogenetic mechanism. This finding aligns with the observed increase in severe cases, suggesting that while the overall WBC count may be lower, the severity of the disease could be attributed to other factors such as inflammatory responses or co-infections that were not captured by the laboratory parameters alone.

4. Mechanistic Interpretations

The discussion attributes epidemiological shifts to "immunity debt" and "immunity

theft." What direct evidence supports these hypotheses in the context of ADV (e.g., serological data on population immunity, viral interference experiments)?

Evidence has documented significant depletion of respiratory syncytial virus (RSV) IgG titers among children under 5 years during the pandemic era (PMID:38009717). However, direct empirical evidence supporting ADV-associated immunological deficit mechanisms—particularly immunity debt from reduced viral exposure or immunity theft through viral interference—remains absent in current peer-reviewed literature. This suggestion points the direction for our future research.

5. Subgroup Analysis

The age distribution shift (1-3 years decreasing, 4-6 years and >6 years increasing) is noted. Could the observed transmission dynamics be attributed to cross-infection within school settings?

We realized that cross-infection within school may play a significant role in the observed transmission dynamics. Our discussion has been expanded to include potential social behaviors, such as increased interactions among older children in school environments, which could facilitate the spread of ADV.

6. Additional Suggestions for Discussion:

1) Regional Variation and Global Trends:

The shift from HAdV-1 to HAdV-3 observed in this study mirrors reports from other countries, including Korea and the United States, indicating a possible global pattern in post-pandemic adenovirus epidemiology. The authors may consider adding a comparative discussion of regional trends and literature data to contextualize the findings (e.g., PMC11375195, PMC7171713).

The prevalence of adenovirus types in other regions is an important point that we have overlooked in our discussion. We have read the related articles including PMC11375195 and PMC7171713 and added a comparative discussion of regional trends and literature data as shown below. "A study from South Korea found that, following the COVID-19 pandemic, the adenovirus subtypes shifted from types 1 and 2 to type 3. However, there are differences in the prevalence of the adenovirus subtypes common in various parts of the world. (13, 15)"

2) Potential Mechanisms Behind Genotype Replacement:

The authors could discuss viral genetic evolution as a potential contributor. A comparative genomic analysis (even theoretical) between HAdV-1 and HAdV-3 could reveal adaptive mutations that may affect transmissibility or immune escape, possibly favoring HAdV-3 in a post-pandemic landscape.

In response to this concern, we have conducted a comparative genomic analysis between HAdV-1 and HAdV-3. Based on current literature, HAdV-1 and HAdV-3 show significant differences in their genomic structure and evolutionary characteristics. HAdV-1 belongs to adenovirus species C, while HAdV-3 belongs to species B. Compared to HAdV-1, the genome of HAdV-3 exhibits greater genetic

variability. For instance, there are 9 amino acid mutations in the neutralizing antigenic epitopes of the hexon gene of HAdV-3. Moreover, specific mutations are also found in the RGD loop and HVR1 region of the penton base of HAdV-3 (PMC7376653, PMC8276179). The genetic variability may favor HAdV-3 in a post-pandemic landscape.

3) Clinical Severity of HAdV-3 Infections:

I recommend that the authors expand their discussion on the observed post-pandemic shift in dominant HAdV genotypes, particularly the replacement of HAdV-1 by HAdV-3, which coincides with a notable rise in the proportion of severe ADV pneumonia cases. This epidemiological transition raises an important question: Is the increase in disease severity directly linked to the genotype shift, or might other post-pandemic factors (e.g., population susceptibility or viral dynamics) also play a role? It has reported that HAdV-7 is more firmly associated with severe outcomes and cytokine storm responses than HAdV-3 in some Chinese children (PMC6327436). So, does HAdV-3 also possess greater intrinsic virulence than HAdV-1? I encourage the authors to reflect on whether genotype-specific pathogenicity could partially account for the increased severity observed and to reference relevant comparative studies, if available.

These additions would enrich the discussion by addressing the broader virological and epidemiological context of the observed genotype shift.

We appreciate the reviewer's suggestions. Current research supports the idea that the HAdV-3 genotype may be a contributing factor to the increase in disease severity (21440492, PMC7376653, PMC8276179), however, there is still insufficient evidence to conclude that it is an independent risk factor. Therefore, a comprehensive multidimensional analysis is required, encompassing host factors, environmental influences, and viral evolution. Additionally, the literature does not directly compare the intrinsic virulence of the two viruses. However, differences can be inferred based on the following clinical and molecular characteristics. HAdV-3 accounts for 37% of hospitalized children, with 28% of those cases staying in the hospital for more than 3 days and 10% requiring intensive care. In contrast, infections with HAdV-1 mostly present with influenza-like symptoms or mild respiratory infections, with no notable proportion of severe cases reported (21440492). The hexon protein of HAdV-3 has multiple mutations in neutralizing antigenic epitopes (such as D207N; T213A; K251R), which might enhance immune evasion and consequently exacerbate inflammatory responses. Additionally, mutations in the RGD loop and HVR1 region of its penton base protein may enhance the virus's ability to invade cells. In comparison, the antigenic genes of HAdV-1 (such as hexon) have not been reported to have similar functional mutations. Its genomic recombination mostly occurs within HAdV species C; moreover, there is no molecular evidence showing an increase in virulence (PMC7376653, PMC8276179).

Reviewer #2 (Comments for the Author):

Thank you for the opportunity to help review the manuscript entitled "Investigating the Please add the Institutional Review Board/human subjects approval information if applicable Retrospective Analysis". Gu et al. investigated how the COVID-19 pandemic affected adenovirus cases, co-infections and severity in children. The study has interesting conclusions, such as identifying a shift from predominating ADV type to type 3, and an increase in overall ADV cases and severity of cases post-lockdown. I have a few comments, suggestions and questions that may help the authors to provide additional information to better support the conclusions.

2022 was not really before the pandemic since the pandemic began in 2020, but more during the pandemic. Please correct the terminology through out to better describe the time periods that were compared.

We already correct the “before the pandemic” to “during the pandemic” throughout the text.

Please add the Institutional Review Board/human subjects approval information if applicable. Overall, more detail needs to be included regarding patient enrollment, consent, storage, testing, etc.

Our research has received approval from the Ethics Board at the Children's Hospital of Soochow University.

About enrollment and consent: All children signed informed consent before admission, agreeing to take part in this clinical study. We enrolled children who tested positive for ADV.

About storage and testing: Children with respiratory symptoms undergo routine nucleic acid testing for pathogens upon admission. The nucleic acid samples of the sputum from children who tested positive for ADV will be stored at -80°C. Once sample collection had been complete, a total of 51 cases were randomly selected for typing tests. This included 26 cases chosen during the pandemic and 25 cases chosen after.

Line 74-78: If these sputum samples were retrospective, how were they stored between hospital admission and genotyping analysis? Or was subtyping performed on all ADV positive specimens as standard of care at time of admission if positive for ADV?

Children with respiratory symptoms undergo routine nucleic acid testing for pathogens upon admission. The nucleic acid samples of the sputum from children who tested positive for ADV will be stored at -80°C. Once sample collection had been complete, a total of 51 cases were randomly selected for typing tests. This included 26 cases chosen during the pandemic and 25 cases chosen after. Due to budget constraints, we were unable to analyze all ADV-positive samples.

Line 81-102 Was the same 13-plex RT-PCR kit/platform described in lines 81-85 used for all 2022 and 2023 patient samples for diagnosis? If not, please list all platforms used as this could affect detection of ADV and also co-infection data. Also, you list

only 10 respiratory pathogens, but say that the kit detects 13? Do you think it would be important to include SARS-CoV-2 detections too? I'm guessing by the dates that most of the children were tested for Covid?

It was the same 13-plex RT-PCR kit used for all 2022 and 2023 patient samples. The 13 respiratory pathogens were including RSV, FluA, FluA-H1N1, FluA-H3N2, FluB, HPIV, ADV, MP, Ch, Boca, HMPV, HRV and HCOV. I think it's important to include SARS-CoV-2 detection. We have confirmed that all ADV-positive children enrolled were tested for Covid and all the test results were negative.

Line 103-108 I suggest also including percentages to report numbers of cases because in 2023 there were almost twice as many specimens tested (similar to the Table 1 results).

It's a good suggestion that we have included percentages to report numbers of cases.

Line 82-83, Table 1: Please define the virus acronyms and clarify why these have different targets than those listed in line 81-102.

It's our mistake. The virus acronyms in lines 82-83 come from the published article. Meanwhile, lines 81-102 use the acronyms from the 13-plex RT-PCR kit instructions. We have now standardized all virus acronyms in the article.

Line 105: does multiple pathogens mean ≥ 2 pathogens other than ADV? Should it be >2 (or ≥ 3) pathogens since ADV single co-infections are 2 pathogens?

Multiple pathogens do mean ≥ 2 pathogens other than ADV. The way we express might have led to a misunderstanding, so we have changed "multiple pathogen (≥ 2 pathogens)" to "multiple pathogen (>2 pathogens)".

Lines 61-70, 94-102: Do you think respiratory virus seasonality affected these results? Some cases spike in different seasons and different years. ADV type 3 has been documented to be one of the most common serotypes detected but it depends on the year and the geographical location.

I think respiratory virus seasonality affected these results. Respiratory viruses have specific peak seasons throughout the year, with cyclical outbreaks occurring every few years. It is worth considering whether the outbreak of ADV-3 following the COVID-19 pandemic coincided with its expected cycle. Moreover, after the COVID-19 pandemic, outbreaks have been observed not only of ADV-3 but also of influenza viruses and *Mycoplasma pneumoniae*. ADV-3 is unlikely to temporally coincide with outbreaks of influenza and *Mycoplasma pneumoniae*. Therefore, we believe that the ADV-3 outbreak remains influenced by the earlier COVID-19 pandemic.

Figure 1 and 2: I suggest clarifying that these figures show the distribution of adenovirus co-infections and not co-infections that don't include ADV (if I'm interpreting it correctly). Additionally, figure one says "before the pandemic" for both

sets of data.

That is a good suggestion. We have revised the figure legend of Figure 1 to 'The proportional distribution of co-infection pathogens with ADV during and after pandemic'. And We have updated the Y-axis label of Figure 2 to 'Pathogens co-infected with ADV'. The "before the pandemic" for both sets of data in Figure 1 is a typographical error. Specifically, the left set of data in Figure 1 should be "during the pandemic," and the right set of data should be "after the pandemic." This has been corrected.

Re: Spectrum00879-25R1 (**Investigating the Impact of the COVID-19 Pandemic on Pediatric Adenovirus Infection Patterns: A Retrospective Analysis**)

Dear Dr. Sheng-Hao Hua:

Thank you for the privilege of reviewing your work. Below you will find my additional comments as well as instructions from the Spectrum editorial office.

Please add Institutional Review Board/human subjects approval information to manuscript text. As well as details regarding patient enrollment, consent, storage, testing, etc.

Terminology was changed from "before the pandemic" to "during the pandemic". However, this may not align with what occurred in the area. According to lines 53-55, large-scale circulation of SARS-CoV-2 did not begin in their area until 2023 when the restrictions were lifted. The authors should consider using other terminology such as "during lockdown" and "after lockdown" or other terms that would better describe the time periods.

The authors present data during a lockdown (2022) and after a lockdown (2023) when COVID-19 infections surged in China (line 57). However, it is difficult to assess the levels of ADV infection without having a baseline. What are the typical number of cases and ADV infections prior to 2020? Multiple years of data would be helpful, for example 2017-2020.

Lines 62-65: The "immune theft" concept pertains to individuals previously infected with SARS-CoV-2. How can "immune theft" be determined in this study without data determining if these children were infected with SARS-CoV-2 in the past (i.e., immune status or previously positive for SARS-CoV-2)? Consider revising this statement and others referring to immunity theft throughout the manuscript.

Lines 303-306: Expand on the ADV subtypes. Including some of the following in the discussion section would be informative: Does ADV subtype vary year to year? Is ADV 3 more severe than other subtypes? Was the shift in South Korea also in pediatrics or adults? Were shifts to ADV 3 also seen in other institutions, locally or abroad? If not, what shifts were seen?

Lines 209-216: The authors state only 51 samples were genotyped yet data in this section shows more than 51 samples being genotyped. Please explain or correct.

Revision Guidelines

Publication Fees: For information on publication fees and which article types are subject to charges, visit our website. If your manuscript is accepted for publication and any fees apply, you will be contacted separately about payment during the production

process; please follow the instructions in that e-mail. Arrangements for payment must be made before your article is published.

Sincerely,
Daniel Ortiz
Editor
Microbiology Spectrum

Please add Institutional Review Board/human subjects approval information to manuscript text. As well as details regarding patient enrollment, consent, storage, testing, etc.

We have added Institutional Review Board approval information to manuscript text as well as details regarding patient enrollment, consent, storage, testing, etc.

Terminology was changed from "before the pandemic" to "during the pandemic". However, this may not align with what occurred in the area. According to lines 53-55, large-scale circulation of SARS-CoV-2 did not begin in their area until 2023 when the restrictions were lifted. The authors should consider using other terminology such as "during lockdown" and "after lockdown" or other terms that would better describe the time periods.

After careful consideration of the terminology, we conclude that "during the lockdown" and "after the lockdown" are the appropriate expressions to describe the time periods.

The authors present data during a lockdown (2022) and after a lockdown (2023) when COVID-19 infections surged in China (line 57). However, it is difficult to assess the levels of ADV infection without having a baseline. What are the typical number of cases and ADV infections prior to 2020? Multiple years of data would be helpful, for example 2017-2020.

According to the figure below, we have analyzed the monthly detection rates of ADV infections from January 2017 to December 2023 to establish a baseline for trend analysis.

Lines 62-65: The "immune theft" concept pertains to individuals previously infected with SARS-CoV-2. How can "immune theft" be determined in this study without data determining if these children were infected with SARS-CoV-2 in the past (i.e., immune status or previously positive for SARS-CoV-2)? Consider revising this statement and others referring to immunity theft throughout the manuscript.

Although studies (PMC10666904, PMC11151613) have shown that more than 90% of children experienced SARS-CoV-2 infection shortly after the end of zero-COVID-19 policy in China, we carefully evaluated the evidence and decided to revise the "immune theft" throughout the manuscript.

Lines 303-306: Expand on the ADV subtypes. Including some of the following in the discussion section would be informative: Does ADV subtype vary year to year? Is ADV 3 more severe than other subtypes? Was the shift in South Korea also in pediatrics or adults? Were shifts to ADV 3 also seen in other institutions, locally or abroad? If not, what shifts were seen?

These studies (PMID:20166176, PMID:11117957) indicate that adenovirus subtype does not vary year to year, but rather follows a cyclical epidemic pattern. Evidence shows that the ADV subtypes 3 and 7 are more likely to cause severe pneumonia among children compared to the ADV subtypes 1, 2, and 6 (PMID:34645218, PMID:36655255). The shift in South Korea was both in pediatrics and adults. The shifts to ADV3 were also seen in Jining, China (PMC11995490). All the aforementioned points have been included in the discussion.

Lines 209-216: The authors state only 51 samples were genotyped yet data in this section shows more than 51 samples being genotyped. Please explain or correct.

We have verified and correct the relevant parts in the article.

Re: Spectrum00879-25R2 (**Investigating the Impact of the COVID-19 Pandemic on Pediatric Adenovirus Infection Patterns: A Retrospective Analysis**)

Dear Dr. Sheng-Hao Hua:

Your manuscript has been accepted, and I am forwarding it to the ASM production staff for publication. Your paper will first be checked to make sure all elements meet the technical requirements. ASM staff will contact you if anything needs to be revised before copyediting and production can begin. Otherwise, you will be notified when your proofs are ready to be viewed.

Sincerely,
Daniel Ortiz
Editor
Microbiology Spectrum